# An ex vivo animal model to study the effect of transverse mechanical loading on skeletal muscle
Marisa Sargent[1], Alastair W. Wark [2], Sarah Day[1] & Arjan Buis [1] ✉

In many populations like wheelchair and prosthetic users, the soft tissue is subject to excessive or repetitive loading, making it prone to Deep Tissue Injury (DTI). To study the skeletal muscle response to physical stress, numerous in vitro and in vivo models exist. Yet, accuracy, variability, and ethical considerations pose significant trade-offs. Here, we present an ex vivo approach to address these limitations and offer additional quantitative information on cellular damage. In this study, skeletal muscle tissue from Sprague Dawley rats was isolated and transversely loaded. Histological analysis and fluorescence staining demonstrated that the setup was suitable to keep the tissue alive throughout the experimental procedure. Mechanically induced cell damage was readily distinguishable through morphological changes and uptake of a membrane impermeable dye. Our comparably simple experimental setup can be adapted to different loading conditions and tissues to assess the cell response to mechanical loading in future studies.

The skeletal muscles in our body usually function to produce movement, stabilise joints, and maintain posture. However, in populations like wheelchair users, bedfast individuals, and prosthetic users, the muscles become a weight-bearing structure. Rather than being subjected to loads in the longitudinal direction, the muscles are transversely compressed between the skeletal system and a support surface like a bed or prosthesis. This can damage the tissues and lead to the development of Deep Tissue Injuries (DTI)[1–4].

The underlying mechanisms of DTI can be investigated on different levels. Since experiments inducing tissue damage on humans are widely considered unethical, cell and animal models are used instead. Starting with the micro-scale, the response of single cells and monolayers to mechanical loading has been explored[5–10]. By increasing the model complexity to basic cell constructs and bioartificial muscles (BAM), the meso-scale response has been documented as well[11–15]. On the macro-scale, researchers induced DTI on murines before recording cellular damage[16–18].

All these model systems can be broadly categorised into two groups: In vitro and in vivo models. While both have contributed to our understanding of mechanical loading effects on cellular dynamics, they involve substantial compromises in crucial methodological aspects, including accuracy, controllability, variability, and ethical considerations (Table 1). Consequently, results are potentially less reproducible, incomplete, or misleading. For example, premature cell development and limited hierarchical structures in in vitro studies lead to morphological and functional deficiencies[19–21], which can impact on the mechanical loading response of cells[9]. Generalisation to the broader clinical context is also limited[17,22], hindering the real-world application of the research.

We therefore propose a new ex vivo skeletal muscle model to study the effect of transverse mechanical loading on cells. This approach offers several distinct advantages. First, unlike single cells and cell constructs, ex vivo tissue has a fully developed structure, which is particularly important when assessing the response of highly hierarchically organised muscle. In contrast to in vivo models, our ex vivo approach also enables the targeted mechanical loading of isolated muscle without interference from extraneous structures like the vascular system. This isolation allows for precise analysis of direct deformation responses, enhancing analytical accuracy. Obtaining ethical approval for working with animal tissue ex vivo is also less complex than applications for in vivo studies, which accelerates the research process while maintaining ethical standards. An additional benefit of the ex vivo system is that similar models are commonly employed in myopathic and exercise-related research[23–26], enabling the comparison of muscle damage studies across different fields. Lastly, the model is adaptable to human biopsy samples, thereby opening the opportunity for clinical translation of research findings.

The aim of this paper is to describe and validate the proposed ex vivo animal model and transverse mechanical loading setup through histology

¹Department of Biomedical Engineering, University of Strathclyde, Glasgow, United Kingdom. ²Department of Pure and Applied Chemistry, Technology and Innovation Centre, University of Strathclyde, Glasgow, United Kingdom. ✉e-mail: arjan.buis@strath.ac.uk

**Table 1 | Advantages and disadvantages of existing pressure injury models**

| Methodological consideration | Example | Tissue injury model system | | |
|---|---|---|---|---|
| | | Single cells | Cell constructs | In vivo animal model |
| Model accuracy | Representation of true hierarchical structure and physiological function | + | ++ | ++++ |
| Controllability of damage mechanism | Separation of direct deformation, ischaemic, reperfusion, and lymphatic damage | +++ | ++ | + |
| Low variability | Environmental and inter-subject variability | ++++ | ++++ | + |
| Ethics | Fulfilment of 3Rs* and ease of ethical procedures | ++++ | ++++ | + |

*Replacement, reduction, refinement.
Scoring key represents how well the model systems realises different methodological aspects with + being the lowest and ++++ the highest score.

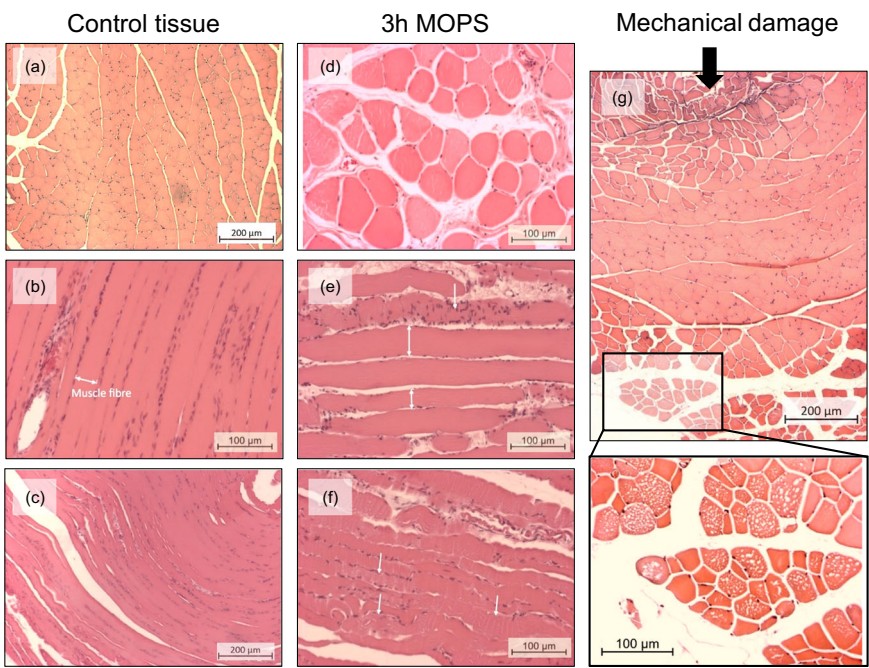

**Fig. 1 | Representative histological images of skeletal muscle tissue.** Negative control samples were directly fixed, processed, and stained with H&E. **a** cross-sectional view of healthy, densely packed muscle fibres with polygonal shape. **b, c** longitudinal view of skeletal muscle fibres in dense, parallel arrangements with only few interstitial spaces; most nuclei located peripherally. 3 h MOPS samples were stored in MOPS-based buffer solution for 3 h before fixation, processing, and staining with H&E. **d** cross-sectional view with rounded cell morphology, swelling, irregular staining, and increased interstitial space. **e** longitudinal view with swollen fibres (compare length of double-headed arrows indicating fibre cross-section); increased interstitial space and accumulation of nuclei (white arrow) **f** longitudinal view showing loss of cross-striation and disorganised fibre structures (white arrows). Mechanical damage samples were transversely indented with up to 60 N, fixed, processed, and stained with H&E. **g** indentation site and direction marked by black arrow; discernible cellular deformation at the impact site as well as increased staining intensity and interstitial spaces; inlet shows magnification of the bottom area highlighting cellular swelling and fibre degeneration. All samples were taken from Sprague Dawley rats, and magnifications are shown in images.

and fluorescence imaging. The feasibility of the model setup was assessed with standard histology, which was also used to describe mechanical damage after static compression qualitatively. Unlike existing methods, we also performed quantitative cell death analysis through fluorescent ProY staining. This allowed us to obtain an objective indication of the damage extent, which is particularly valuable for comparative studies. Our vision for introducing this ex vivo model system is to complement existing approaches and thereby create a more comprehensive understanding of the effect of physical stress on skeletal muscle.

## Results
### Suitability of skeletal muscle storage in MOPS-based buffer solution
The cross-sectional view of histology slides from both the control group and directly processed muscles displayed the typical polygonal appearance of skeletal muscle cells for most samples (Fig. 1a). Only a few interstitial spaces

were present, and nuclei were placed peripherally. Staining intensity was uniform across the samples. In some samples, peripheral cells displayed signs of damage, including increased interstitial spaces and rounded morphologies. This damage might be the result of chemical fixation[27], or was introduced during the dissection process. Compared with control samples, small areas with minor cell damage were visible in the samples stored in MOPS solution ((3-(N-morpholino) propanesulfonic acid) for 3 h. The damage was marked by increased interstitial spaces, varying cell size and staining intensity, rounded morphologies, and centralised nuclei (Fig. 1d).

Looking at control samples in a longitudinal view, cross-striation was visible as well as the structural arrangement of cells in long fibres with peripheral nuclei (Fig. 1b, c). These arrangements were sometimes disorganised in MOPS-stored samples (Fig. 1e, f), especially when samples were not pinned out to prevent involuntary contractions. Additionally, striation was partially lost and structural damage apparent. Some fibres also showed signs of swelling and multiple internal nuclei.

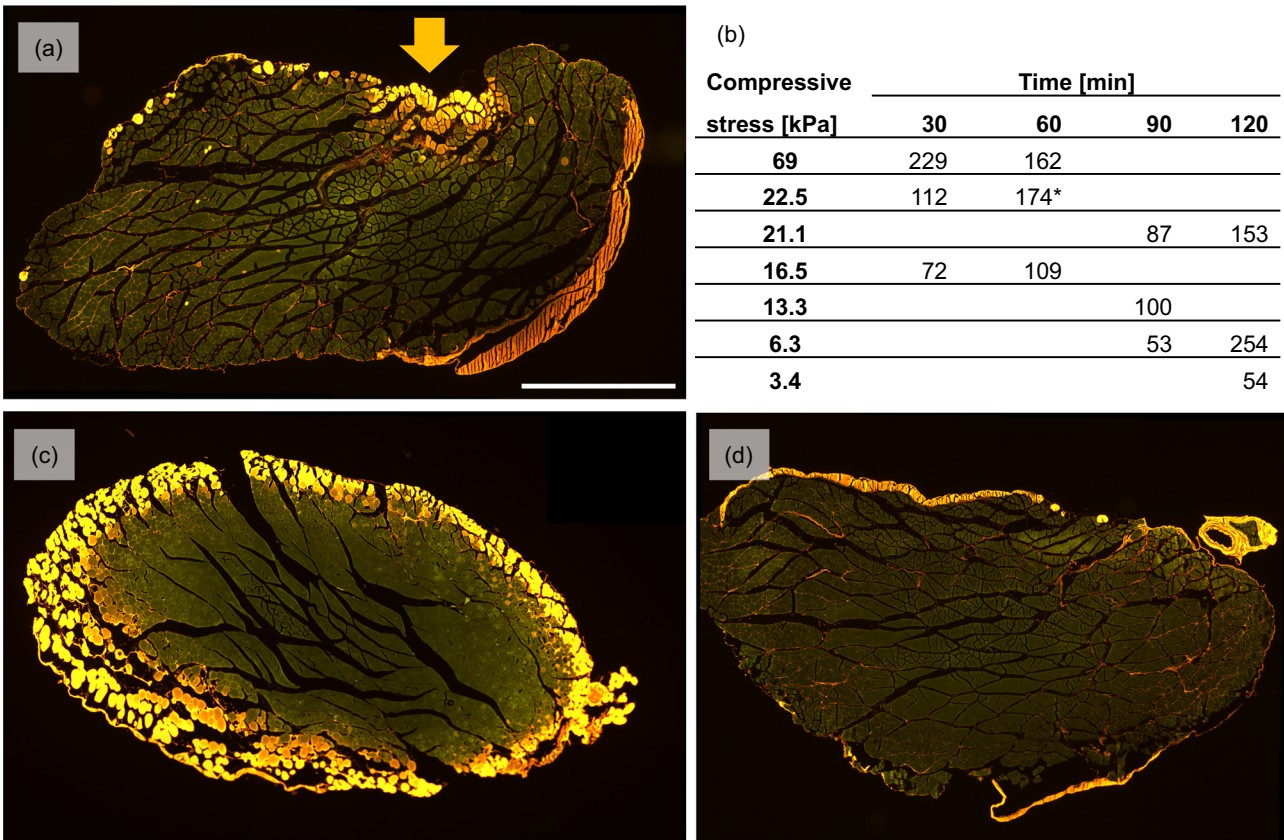

(a)

(b)

| Compressive stress [kPa] | Time [min] | | | |
|---|---|---|---|---|
| | 30 | 60 | 90 | 120 |
| 69 | 229 | 162 | | |
| 22.5 | 112 | 174* | | |
| 21.1 | | | 87 | 153 |
| 16.5 | 72 | 109 | | |
| 13.3 | | | 100 | |
| 6.3 | | | 53 | 254 |
| 3.4 | | | | 54 |

(c)

(d)

**Fig. 2 | Representative images of cell death following mechanical loading.** All images represent fluorescent labelling of dead skeletal muscle cells with Procion Yellow MX4R. Viable cells have a green autofluorescence signal, dead cells are marked by yellow staining. Images were obtained with a FITC filter (Ex 482/35, Em 536/40). Scale bar 1000 µm. **a** The sample was transversely compressed (yellow arrow) for 60 min with a mass of 11 g, creating a compressive stress of 22.5 kPa. **b** The table displays the average number of dead cells in skeletal muscle following different mechanical loading protocols. Each pressure-duration combination was tested on one skeletal muscle sample from Sprague–Dawley rats, and the cell death count averaged across six cross-sectional images. The time frame for the pressure application ranged from 30 min–120 min, the internal compressive stress ranged from 3.4 kPa to 69 kPa. The asterix (*) indicates the average cell death count for the sample displayed in image (**a**). **c** Positive control sample showing dead cells in the outer layers of the muscle sample after chemical lysis with Triton X-100 for 2 h. **d** Negative control sample showing very minimal cell death after storage in MOPS buffer for 1 h without any intervention.

## Basic histological assessment of cellular damage in mechanically indented skeletal muscle tissue

In mechanically damaged, Haematoxylin and Eosin (H&E) stained samples, the indentation site was clearly distinguishable (Fig. 1g). In both longitudinal and cross-sectional slides, uneven staining, increased interstitial spaces, inflammatory cell infiltration, and swelling of damaged fibres was visible. Notably, the majority of the tissue damage was located close to and opposite to the indentation site whilst the middle regions of the muscle seemed intact. This is congruent with the results from fluorescence imaging as well other studies that reported a damage propagation away from the loaded region[28,29].

## Quantitative cell-death analysis in fluorescently stained and mechanically indented skeletal muscle tissue

The average cell death count obtained from statically indented and fluorescently labelled samples ranged from $n = 53$ to $n = 254$ (Fig. 2a, b). The lowest compressive stress for each duration corresponded with the lowest number of dead cells. For medium and high-level stress, however, cell death numbers were similar with $n = 160 \pm 70.56$ at medium and $n = 158 \pm 58.09$ at high stress levels. Unexpectedly, at three out of four time points, the number of dead cells after high stress was below that of the medium stress level. A longer duration of stress application on the other hand led to more dead cells for all but one stress level (69 kPa for 30–60 min).

To verify the stains specificity, positive and negative controls were analysed. Samples treated with Triton X-100 displayed ring-shaped layers of dead cells in the periphery (Fig. 2c). In negative control samples without intervention, no cell death was apparent (Fig. 2d). The absence of excessive cell death in the negative control samples should also be interpreted in the context of their long-term storage in MOPS buffer. The results confirm that despite the cellular re-organisation documented in the basic histology study, cells remained viable throughout the course of the experiment.

## Stress-time cell death threshold for ex vivo skeletal muscle tissue

A stress-time plot was generated with a differentiation between the "cell death" and "no damage" datapoints (Fig. 3a). From this, sigmoid stress-threshold functions were determined for each category with the parameters for the graph defined in Fig. 3. Both "cell death" and "no damage" functions encapsulate a region of uncertainty where the fate of a cell is not clearly destined. One outlier was observed at 90 min loading with 21.1 kPa. The sample was classified as "no damage" despite the compressive stress being above the proposed threshold function. This might be related to human error as the staining procedure for this sample deviated from the protocol followed for the other samples, i.e., the sample was not "flipped" to ensure sufficient contact between the sample and the staining solution. Another potential explanation is biological variability. However, with only one sample analysed for each load-time-combination to minimise animal usage, no definitive conclusion could be drawn. With only a few data points available, the outlier was excluded from the threshold curve analysis to avoid skewed results.

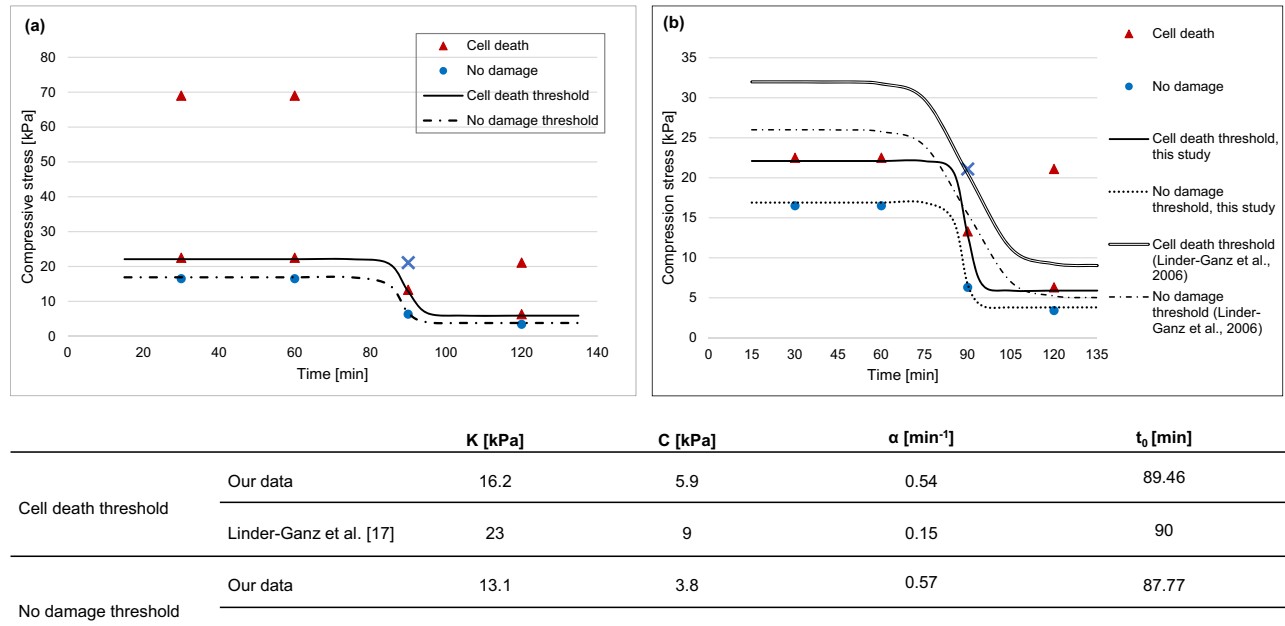

| | | K [kPa] | C [kPa] | α [min⁻¹] | t₀ [min] |
|---|---|---|---|---|---|
| Cell death threshold | Our data | 16.2 | 5.9 | 0.54 | 89.46 |
| | Linder-Ganz et al. [17] | 23 | 9 | 0.15 | 90 |
| No damage threshold | Our data | 13.1 | 3.8 | 0.57 | 87.77 |
| | Linder-Ganz et al. [17] | 21 | 5 | 0.15 | 90 |

**Fig. 3 | Static stress-time cell death thresholds for skeletal muscle tissue. a** Markers indicate data obtained from indentation of ex vivo skeletal muscle with triangles signalling "cell death", circles marking "no damage" situations, as quantified through fluorescence dead cell staining. One outlier was observed at 90 min, 21.1 kPa loading, marked with a blue cross. The cell death number at this point classified the sample as "no damage" despite the compressive stress being above the proposed threshold. **b** Comparison of stress-time cell death thresholds from this ex vivo and Linder-Ganz et al.'s in vivo studies[17]. The threshold curves from in vivo studies have a similar time step as those fitted to the data from this study but differ in stress magnitude and range, as well as the slope of the sigmoid step. The corresponding parameters of the sigmoid Boltzmann functions (Eq 1) for the thresholds are shown in the table with $K$ as the range of the function, $α$ as slope coefficient, $t_0$ as time at midrange, and $C$ as minimum compressive stress asymptote.

The conformity of the threshold functions with the obtained data can be seen in more detail in Fig. 3a. The experimental data visually overlapped well with the determined threshold levels. The sums of the squared residuals (SSE) were calculated in Minitab to measure unexplained variability by the regression. The low values for both threshold curves ($SSE_{death} = 0.72$, $SSE_{no\ damage} = 0.62$) support the observed good fit.

When comparing the results of this study to those of Linder-Ganz et al.[17], the parameters of the threshold functions were different (Fig. 3b). The loading level of both cell death and no damage scenarios was lower in the current study, with the stress range $K$ reduced by 6.8 kPa, the minimum stress $C$ by 3.1 kPa. On the contrary, the slope coefficient was higher in the current study with $α_{death} = 0.54\ min^{-1}$ and $α_{no\ damage} = 0.57\ min^{-1}$ compared to $α = 0.15\ min^{-1}$ in Linder-Ganz et al.'s results, leading to a steeper sigmoid step. The time at midrange was similar across both studies with $t_0 = 90\ min$ for Linder-Ganz et al.'s study, and $t_0 = 89.5\ min$ and $t_0 = 87.8\ min$ for the "cell death" and "no damage" thresholds of this experiment.

## Discussion

A new ex vivo model of skeletal muscle was proposed together with a fluorescent staining protocol to quantify sarcolemmal damage following transverse mechanical loading. Basic histology with H&E staining as well as fluorescently stained negative control samples showed that sample storage in MOPS buffer seemed to keep the skeletal muscle tissue reasonably stable during the experiments when the time frame was kept to 3.5 h. However, the lack of oxygenation may introduce minor cell damage. To minimise the potential influence of storage-related cell death on the mechanical cell death measurements, comparison of each intervention group with a control that was kept in identical environmental conditions is paramount. We, therefore, physically divided each sample after the intervention: areas at the indentation site were allocated to the "intervention" group; untouched areas in the periphery served as control sites.

When comparing the mechanical damage and control group, clear indicators for cellular damage were visible in the H&E-stained tissue. However, comparing the extent of cellular damage after different mechanical loading scenarios might be more difficult and at higher risk of bias with a qualitative approach. Accordingly, fluorescent staining was employed as an alternative method and used for validation of the model system.

The performance of the ex vivo model was measured against an established in vivo model by Linder-Ganz et al.[17]. They identified a sigmoid pressure-time cell death threshold, characterising the critical turning point where cells transit from "no damage" to "cell death". Both models recorded a similar sigmoid-shaped stress-time cell death that could differentiate undamaged and damaged samples. For the current study, the proximity of the results at the 90 min loading duration that differentiate between "cell death" and "no damage" provides a level of confidence in the sigmoid shape. However, sample numbers were low, with only one muscle per pressure-time-combination, leading to overfitting of the sigmoid threshold curve. The comparison between different model systems should therefore be interpreted with caution as the sigmoidal response proposed in this paper is not a unique solution. For future experiments, more samples should be included to account for biological variability and improve the significance of results.

Other researchers have also defined sigmoid threshold levels for skeletal muscle but with different model systems. Yao et al.[9] used monolayers of myoblasts, Gefen et al.[13] performed experiments on BAM. However, the characteristics of the threshold curves differed significantly between the model systems. For example, myoblast monolayers seem most resistant to prolonged loading, followed by BAM and biological tissue. The sigmoid step time (point at which the decline of cell tolerance is at its maximum) was reached after 160 min for myoblasts[9], 116 min for BAM[13], and 90 min for biological tissue (Fig. 3).

This and other differences observed in stress magnitude tolerance and stress endurance might be explained by differences in hierarchical structure

and cell morphology. In natural muscle, cells are arranged anisotropically in a fibrous pattern surrounded by connective tissue. When compressive stress is applied, the fibrous arrangement of cells attached to sheets of connective tissue causes additional tensile forces[30], thereby increasing the mechanical load on the cell membrane. Over time, the structural breakdown of the cytoskeleton and cellular membrane might accelerate. In comparison, the less mature connective tissue network of BAM and lack of fiber arrangement and longitudinal cell shapes in monolayers make them less prone to tensile stress. Apart from the mechanical environment, the level of cellular maturity might play a role. Genetically modified cells like C2C12 myoblasts do not fully represent the structures and physiological processes that a mature, primary cell displays[9]. The mechanical stress response, therefore, might be different. Additionally, not only the biological material used in each of the studies but also the experimental setup, processing, and analysis steps to assess cellular damage differed. The comparison of absolute values should therefore be treated with caution.

Nevertheless, the response of the ex vivo model to static compression seems to closely resemble that of in vivo tissue, despite the lack of a functional blood and lymphatic supply. Other researchers also found that ischaemic damage has a later onset than direct deformation damage[14], making it less relevant for comparably short experimentation times (<4 h). It is, therefore, likely that for short-term mechanical loading experiments, like this study, the developed ex vivo model evokes a similar soft tissue response to direct stress as an in vivo model. This confirms that the proposed ex vivo model and image analysis workflow presents a compelling alternative to explore the influence of mechanical loading on skeletal muscle health.

One main difference between the discussed threshold studies is the way the threshold is defined. Linder-Ganz et al.[17] determined cellular damage semi-qualitatively, assigning samples to "cell death" or "no damage" groups. Gefen et al.[13] used the relationship between the geometry of the indenter and the location of Propidium Iodide-positive cells. While these approaches are suitable for defining categorical threshold levels, they make it difficult to assess the influence of loading characteristics like magnitude and duration more specifically.

The developed ex vivo approach allows to obtain additional quantitative information (Fig. 2). In relation to the relevance of the pressure magnitude on skeletal muscle health, one might expect that the higher the pressure, the higher the number of dead cells. However, the results of this study did not support this hypothesis. On the contrary, at three out of four time points, the highest applied pressure produced only the second highest cell death counts. Kosiak[31] reported a similar soft tissue response for an in vivo model. Like in this validation study, once cell damage was visible, its extent remained relatively constant with increasing pressure magnitudes. The duration of direct stress application, on the other hand, seems to influence the extent of cellular damage; the longer the load application, the more damage was visible at a constant compressive stress in all but one case. Other researchers observed comparable results in an in vivo[32] setup in rats as well as in vitro studies[9]: Once an initial threshold was exceeded, tissue damage spread with increased loading duration.

Overall, the results emphasise the importance of obtaining quantitative data in addition to defining "cell death" and "no damage" regions as it provides more information on the development of DTI. The similarities in the cellular response between the quantitative in vivo and in vitro studies and the performed ex vivo study are also another indication for the validity of the proposed setup.

One interesting observation from the image analysis was that cellular damage occurred mainly near the contact between the muscle and indenter. A possible explanation is restricted nutrient supply caused by the impermeable indenter. Although we did not validate this theory as part of our experiments, other studies showed no effect of contact between an impermeable indenter and cells[33]. Moreover, the observed cell death was in some cases located deeper within the tissue without extending to the interface. Instead, a heterogeneous stress field might play a role. A macro-level Finite Element (FE) model such as that used in this study can only describe the mechanical environment within the tissue in general. This might differ from the environment experienced by single cells because of the complex anisotropic and heterogenous structure of skeletal muscle. It is hence possible that the compressive stress magnitude at cellular level differs from the calculated values, or additional tensile and shear stresses are present. Further multi-scale FE modelling would be necessary to explore this theory.

To summarise, this study presents a series of strategies to process and image ex vivo skeletal muscle tissue after mechanical stress situations with a potential for quantitative cell death analysis. The foundation is ex vivo murine tissue that is subjected to transverse mechanical loading before being stained and processed for imaging with an epifluorescence microscope. With ProY, we identified a well-established marker for skeletal muscle damage and verified its capability to highlight sarcolemmal damage in mechanically stressed tissue. It is compatible with standard formalin fixation and paraffin embedding (FFPE) as well as with cryosectioning, as other studies demonstrated[34–36]. The staining procedure is rather simple, compared with immunofluorescence protocols for example, and yet specific enough to indicate membrane damage for quantitative analysis. Nevertheless, other staining or microscopy techniques can also be integrated.

The model was validated by comparison with an existing in vivo model[17] and discussed in the context of other threshold studies performed in an in vitro environment[9,13]. The ex vivo model thereby combines benefits of in vivo and in vitro experiments, like the hierarchical structure, intact function, and a controllable environment. Consequently, different environmental conditions mimicking ischaemia or reperfusion[9,13] could be created. The samples are also well-suited for transverse mechanical loading studies, yet not restricted to this form of setup. The compatibility with most mechanical testing equipment means that other conditions such as dynamic compression can be applied, as done by our team in further studies. Lastly, by adjusting the loading setup, not only murine muscle but fresh human biopsies could be tested. This could give an indication for the translatability of murine test results to human studies and bridge the gap from the lab to the clinical environment. Overall, the developed ex vivo animal model will hopefully provide a basis for standardised research not only for pressure injury but adjacent areas.

## Methods
### Collection and preparation of murine skeletal muscle tissue
The soleus (SOL) and extensor digitorum longus (EDL) muscles were isolated from the hindlimbs of 6–7 weeks old male Sprague-Dawley rats within an hour of euthanisation. The muscles had an average weight of 0.11 ± 0.02 g (SOL) and 0.12 ± 0.02 g (EDL). Their cross-sectional diameter was approximately 2 mm. Both muscles are easy to access and dissect and have a low pennation angle[37], which makes it easy to achieve a constant direction of mechanical loading in relation to the fibre orientation and to produce cross-sectional muscle slices for microscopy. Following dissection, the muscles were pinned out in Petri dishes covered with medical grade silicon. This prevented fibre distortion from involuntary contractions that was otherwise observed. To enhance the viability of excised muscle tissue during the experiment, it was submerged in MOPS-based buffer solution and kept at ambient temperature (22 °C). The physiological buffer was prepared from 145 mM sodium chloride, 2 mM MOPS (3-(N-morpholino) propanesulfonic acid), 4.7 mM potassium chloride, 1.2 mM monosodium phosphate, 1.2 mM magnesium chloride, 5 mM glucose, 0.2 mM EDTA, 2 mM sodium pyruvate, and 2 mM calcium chloride, all acquired from Sigma Aldrich, St. Louis, and adjusted to pH 7.4.

All experiments were carried out on freshly dissected tissue from animals not subject to any other treatments. The animals were housed at a licensed establishment (Establishment Licence number X56B4FB08) in accordance with the code of practice for the housing and care of animals bred, supplied or used for scientific purposes. Killing was in accordance with UK regulations (Animals (Scientific Procedures) Act 1986, revised under European Directive 2010/63/EU). Male Sprague-Dawley rats were killed by trained technicians with an intraperitoneal overdose of sodium

**Fig. 4 | Setup for transverse mechanical loading.**
**a** 3D printed static loading setup. Consists of 3D-printed loading rig with plunger that has a steel rod with flat-ended round indenter attached to it ($r = 1$ mm). The construct fits on top of a standard 100 mm Petri dish that was filled with medical-grade silicone to facilitate pinning of muscle specimen and prevent slipping of the sample. This setup was used for loads ≥11 N. **b** CAD model of static loading device that allows for reproducible 3D printing and customisation. **c** Static loading setup with Bose Electroforce 3100. Pinned out muscle sample is inserted at the bottom of the machine. An indenter is lowered onto the sample. This setup was used for loads <11 N. **d** Close-up of loading setup within Bose Electroforce 3100.

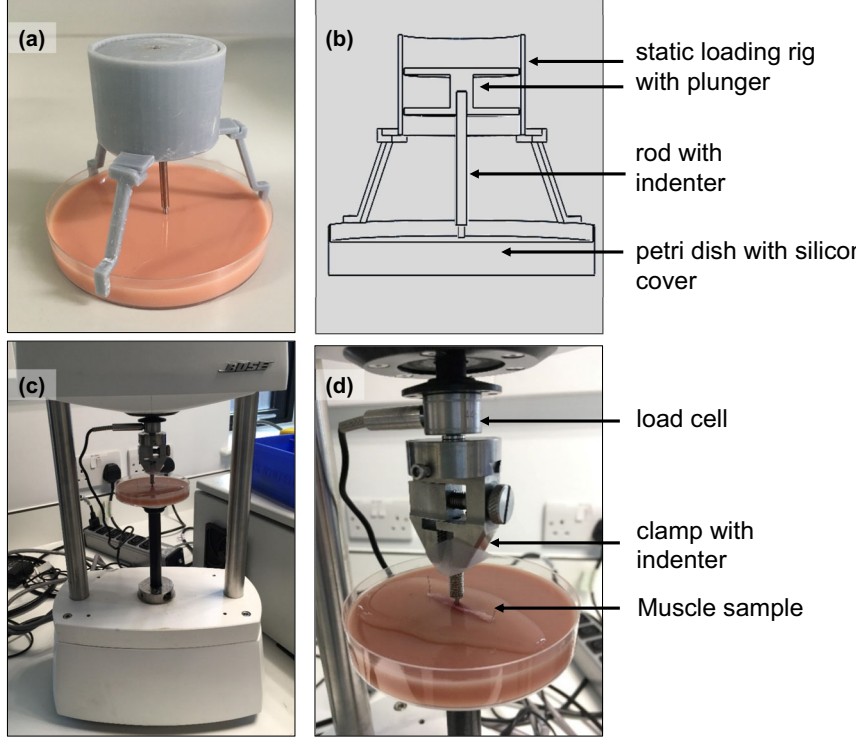

pentobarbital (Euthatal, 200 mg kg$^{-1}$). The 3 R principles of replacement, reduction, and refinement[38] were followed wherever possible. Partial replacement was ensured by using tissues taken from animals without conducting previous scientific experiments that could cause suffering. Reduction was also achieved by taking tissue from rats that have been assigned for other non-muscle-based research and by using four muscles per animal to minimise the number of animals. Additionally, for all indentation experiments, each muscle was divided into indentation and control areas (outside of indentation zone), which were analysed separately.

#### Mechanical loading setup and instrumentation

To inflict mechanical damage to skeletal muscle, transverse compression was applied by lowering a flat, circular indenter ($r = 1$ mm) onto the samples. The indenter was either part of a customised loading rig or attached to a Bose Electroforce 3100 (TA Instruments) to allow for parallel processing of four muscle samples at once. The custom rig was developed with CAD software (SolidWorks 2019) for 3D printing (Fig. 4a, b). The steel indenter had a mass of 11 g to deliver a direct stress of at least 30 kPa to the muscle. This is equal to the pressure resulting in cellular damage after static compression for 30 min[17]. For loads >11 N, weight was added to the top of the indenter.

The Bose Electroforce was used for loads <11 N. It was fitted with a load cell (22 N) connected to a flat steel indenter (Fig. 4c, d). The system was allowed to stabilise for 30 min prior to usage. The Petri dishes with samples were inserted on the bottom plate and raised towards the indenter. To initiate contact, the top mover was activated to bridge the remaining distance until a contact force of $-+30.01$ N was established.

#### Finite element analysis of transverse mechanical indentation

The pressure externally applied by the indenter might not match the compressive stress within the sample. Accordingly, finite element analysis (FEA) was performed in Ansys (Workbench 2021 R2) to determine the mechanical conditions within the muscle samples for each pressure application. An axisymmetric model of the indentation setup was designed with rectangles representing each body—indenter, muscle, silicone—according to real-life measurements.

The indenter and silicone were both assigned linear elastic material properties ($E_i = 200$ GPa, $v_i = 0.3$; $E_S = 3.42$ GPa, $v_S = 0.495$) whereas the muscle was described by a first-order Ogden model ($\mu = 15.6$ kPa, $\alpha = 21.4$) with Prony series expansion ($\delta = 0.549$, $\tau = 6.01$ s)[39]. The muscle model is based on experimental results from transverse mechanical compression of rat skeletal muscle tissue, making it very relevant to this study. Additionally, Linder-Ganz et al.[17] used the same model to predict the internal loading conditions in compressed skeletal muscle in their in vivo experiments, which served as validation for our new ex vivo model. By using the same FE model, we ensured better comparability between both studies.

The chosen model indenter was meshed through general face meshing with quadrilateral elements, the mesh for the muscle and silicone had a defined element size of $2.5 \times 10^{-9}$ m$^2$, based on results of a mesh convergence study. The contact between the indenter and muscle was assumed to be frictionless; the contact between the muscle and silicone layer allowed sliding but prohibited separation. Displacement restrictions in the X-direction were placed on the axis of symmetry, and the bottom of the silicone layer was treated like a fixed support.

A direct stress ("pressure" in Ansys) was then applied to the top line of the indenter, according to the loading scheme described in Fig. 5c. The internal compressive stress was predicted to be 32% below the external interface pressure ($R^2 = 99.76\%$, Fig. 5d). The stress-field in general was mostly uniform across the indentation zone. Localised stress concentrations were observed underneath the indenter edge. However, the majority of the stress field underneath the indenter was homogeneous. Based on these results, computed values of the average internal stress in the homogeneous area (Fig. 5a, b) were used instead of external direct stress to interpret data from the indentation study.

#### Standard histology for feasibility study and qualitative analysis of mechanically damaged ex vivo tissue

To test the feasibility of the ex vivo setup, skeletal muscle samples were isolated from Sprague-Dawley rats and either directly fixed and processed ($n = 6$) or stored in MOPS for 3 h at 22 °C prior to processing ($n = 5$) for comparison. Additionally, $n = 9$ muscles were mechanically damaged through transverse indentation with a force of 11–60 N. Muscles from the

**Fig. 5 | Representative finite element models of the ex vivo indentation experiments. a, b** Images demonstrate relationship between external pressure application (5 kPa and 103 kPa) and average compressive stress within the skeletal muscle layer (3.4 kPa and 69.1 kPa) when statically compressed with a flat indenter into a silicone layer (not shown in images). **c** Overview of static compressive loading protocol. Compressive loads ranging from 0.02–0.32 N (principal compressive stress 3–68 kPa) were applied to skeletal muscle samples for durations between 30 and 120 min. Each compression-duration-combination represents one sample and was chosen based on the pressure-cell death threshold defined by Linder-Ganz et al.[17]. **d** Difference between external direct stress and mean internal muscle stress applied by indentation experiment. Strong linear relationship ($R^2 = 99.76\%$) between externally applied direct stress and difference between direct and internal stress as estimated through finite element analysis.

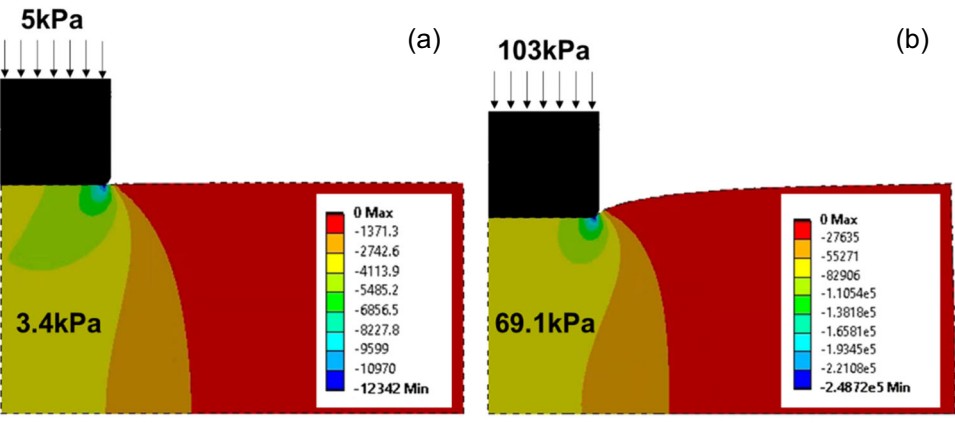

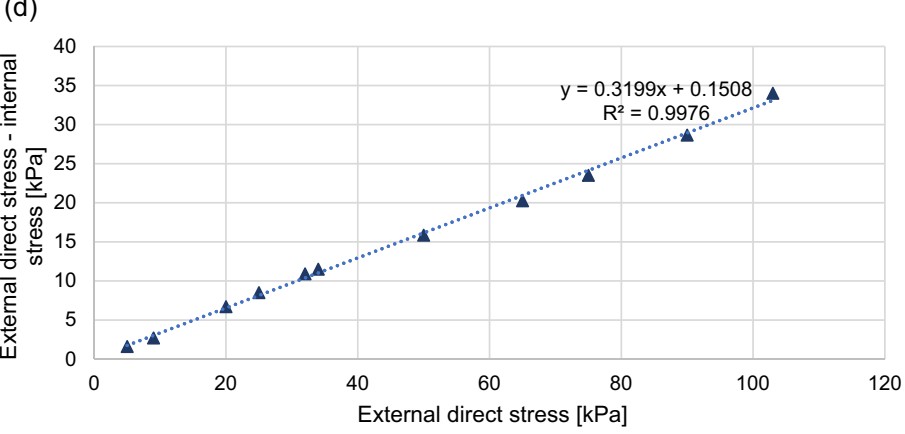

(c)

| | 30 min | 60 min | 90 min | 120 min |
|---|---|---|---|---|
| | 0.32 N / 68 kPa | 0.32 N / 68 kPa | 0.10 N / 21 kPa | 0.10 N / 21 kPa |
| | 0.11 N / 22 kPa | 0.11 N / 22 kPa | 0.06 N / 13 kPa | 0.03 N / 6 kPa |
| | 0.08 N / 17 kPa | 0.08 N / 17 kPa | 0.03 N / 6 kPa | 0.02 N / 3 kPa |

contralateral limb served as negative controls ($n = 9$). They were held in the same conditions for the same time but without any load application.

Next, the samples were FFPE. Each muscle was fixed in 10% neutral buffered formalin (NBF, Sigma Aldrich) for 28 h before being processed overnight (Thermo Shandon Citadel 1000) in graded alcohol (Fisher Scientific), Histoclear (National Diagnostics) and Paraffin (Raymond Fisher Scientific). The tissues were then embedded in wax (BDH Laboratory Supplies Paramat) and cut into 3–4 µm thick cross-sectional or longitudinal slices on a microtome (Leica RM2125RTF).

A standard Haematoxylin and Eosin (H&E) protocol was followed to visualise the microanatomy of the tissues. The samples were submerged in a series of Histoclear, graded EtOH, Harris' Haematoxylin (Sigma Aldrich), tab water, Acid Alcohol (Sigma Aldrich), and Eosin Y (Sigma Aldrich). Once stained, the samples were mounted with DPX (Sigma Aldrich).

Basic histology samples were imaged either with a Carl Zeiss microscope (Axioimager Z1) at ×10 and ×20 magnification or a Motic BA210 (with Moticam 10+ Camera and Motic images plus 3.0 software) at ×10, ×40, and ×100 magnification. All images were analysed qualitatively with Zen 3.1 (Carl Zeiss) and FIJI/ImageJ (Vers. 1.53 s, NIH).

### Fluorescence staining for quantitative analysis of mechanically damaged ex vivo tissue
To validate the ex vivo model system, static mechanical compression was applied. The resulting cell death was measured and compared to an in vivo

study by Linder-Ganz et al.[17] as well as in vitro studies by Gefen et al.[13] and Yao et al.[9]. The static compressive stress ranged from 3–68 kPa and was applied for 30–120 min (Fig. 5c). For each duration, three pressure values were chosen that represented the "cell death" (=above threshold), "threshold" (=transition zone), or "no cell death" (=below threshold) region defined by Linder-Ganz et al.[17]. Each pressure-duration-combination was tested on one sample and the maximum experimental duration was restricted to 3.5 h to ensure tissue viability. Areas in the periphery of the damaged muscles were used as negative controls. For positive controls, we induced chemical damage through Triton X-100 treatment, which results in cell lysis. Samples were submerged in 0.1% Triton X-100 (Sigma Aldrich, St. Louis) for 30 minutes.

All samples were fluorescently stained with Procion yellow MX4R, also known as ProY or formerly Procion Orange. This membrane impermeable, fluorescent dye has been used to visualise sarcolemmal damage in ex vivo tissue by binding to the cytosol[40–43]. Following the experimental procedure, samples were submerged in 0.1% ProY (Sigma Aldrich, St. Louis) in PBS for 30 min before being washed in PBS 3 × 5 min and FFPE processed.

### Fluorescence microscopy and image analysis
Fluorescently stained slides were imaged on a Leica Microsystems SP8 confocal microscope in widefield-mode. The system consisted of a Leica DMi8 inverted microscope with images acquired via a ×10 objective (HC PL APO CS2, NA 0.4) and a DFC7000T CCD camera. For each mechanically

damaged and control sample, $n = 6$ cross-sections were imaged with 10x magnification through a FITC filter (Ex 450–490 nm, DC: 510, Em 515 nm LP), followed by a Y5 filter (Ex 590-650 nm, DC: 660, Em 662-738 nm). Full cross-sectional images were composed through automated tiling and stitching with the Leica LAS X Navigator software (Leica Application Suite X V.3.1.5.16308).

All images were imported into FIJI/ImageJ and converted to 8-bit greyscale images before an established analysis workflow was performed. In short, all images were background subtracted and normalised before further processing. For control tissue sections, the combined average intensity was determined for each sample. This value was set as threshold to mask unlabelled fibres in sections from the indented areas. Following segmentation with Cellpose[44] to separate individual myofibers, the number of dye-positive cells in the indented area was counted for each cross-section and averaged across each sample.

### Definition of stress-time cell death thresholds for ex vivo skeletal muscle tissue

To validate the ex vivo model, a stress-time cell death threshold was established. However, samples could not simply be classified as "damaged" when ProY-positive cells were detected, as all cross-sections had cell death counts of ≥53. Those dead cells were not necessarily associated with mechanical loading. Possible other sources are naturally occurring cell death, damage from sample handling, or storage without oxygenation. A baseline-level above which a cross-section is classified as damaged therefore had to be defined.

The differentiation between "cell death" and "no damage" was based around damage distribution. In samples with high cell death counts, damaged cells were arranged in a centralised pattern, which was also observed by other researchers in similar experiments[18,28,29]. In comparison, samples with low dead cell counts were missing a distinctive pattern with dead cells scattered throughout the samples. It is therefore likely that a centralised damage distribution is related to mechanical damage from the indentation.

Based on this observation, all processed images were viewed to determine whether a distinctive damage pattern was visible. If one or more of the sections belonging to the same sample displayed a localised damage distribution, the whole sample was categorised as "cell death". From this, the lowest average number of dead cells in damaged samples was defined as baseline of $n = 110$, above which samples were categorised as "cell death", and below as "no damage".

The results from the categorisation of samples into "cell death" and "no death" groups were combined with the internal stress values calculated through FEA into a stress-time plot. From this, stress-threshold functions for samples categorised as "cell death" and "no damage" were compiled. Their shape was estimated by four-parameter decreasing single-step Boltzmann-Type sigmoid functions, which have been introduced by Linder-Ganz et al.[17] and Gefen et al.[13] to define muscle-damage-thresholds in in vivo and in vitro studies:

$$\sigma_{compressive} = \frac{K}{1 + e^{\alpha(t-t_0)}} + C \qquad (1)$$

where $\sigma$ is the compressive stress, $K$ [kPa] is the range of the function, $\alpha$ [min$^{-1}$] is the "slope coefficient"[45], $t_0$ [min] is the time at midrange, and $C$ [kPa] represents the minimum compressive stress asymptote. Each dataset was analysed in Minitab (Vers. 19.2020.1), according to the least-square-method. Parameter estimates for $K$ (maximum compressive stress minus minimum compressive stress) and $C$ (minimum compressive stress) were calculated from the dataset, $\alpha$ and $t_0$ were taken from Linder-Ganz et al.'s study[17] (Eqs. 2 and 3).

$$\sigma_{celldeath} = \frac{16.2 kPa}{1 + e^{0.5\,\mathrm{min}^{-1}(t-90\,\mathrm{min})}} + 5.5 \qquad (2)$$

$$\sigma_{nodamage} = \frac{13.1 kPa}{1 + e^{0.5\,\mathrm{min}^{-1}(t-90\,\mathrm{min})}} + 4.2 \qquad (3)$$

To adapt a conservative approach to threshold estimates, the $C$ parameter of the "cell death" graph was restricted to $C < 6$ to be below all "cell death" datapoints and that of the "no death" graph to $C > 3.7$ to be above all "no death" datapoints. The resulting threshold curves were assessed for their fit and compared with those from Linder-Ganz et al.'s study[17].

### Reporting summary

Further information on research design is available in the Nature Portfolio Reporting Summary linked to this article.

### Data availability

The authors declare that the data supporting the findings of this study are representative of a series of measurements to ensure robustness and reliability. Should any raw data files be needed in another format, they are available from the corresponding author upon request.

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

## Acknowledgements

This work was supported by the Engineering and Physical Sciences Research Council, EPSRC (grant number EP/S02249X/1).

## Author contributions

M.S.: conceptualisation, methodology, formal analysis, investigation, writing—original draft, review & editing, visualisation. A.W.: Methodology, Validation, Investigation, Resources, Writing—Review & Editing. S.D.: conceptualisation, supervision. A.B.: conceptualisation, methodology, writing—review & editing, supervision, funding acquisition.

## Competing interests

The authors declare no competing interests.
