## [Peer Review File · Communications Biology]

Reviewers' comments:

Reviewer #1 (Remarks to the Author):

The study "An ex vivo animal model to study the effect of transverse mechanical loading on skeletal muscle" is in accordance with scientific standards. It extends the knowledge of changes in the musculature with regard to mechanical loading.

General comments:

The study is logically structured and comprehensible. However, I would like to see a more detailed overview of the animals used and the muscles taken from them. This concerns in particular the morphometric data regarding the animals, such as sex, weight, age. In general, I would also like to see more of the original data provided.

Specific comments:

Line 71: Please specify SD's.

Line 79: Please introduce MOPS.

Line 120: Here you use the Ogden model. Would there not be a material, that takes the anisometry of the musculature into account, more suitable? <https://doi.org/10.1016/j.ijsostr.2015.11.008>

Fig.2d: The graph should be filled with data points between 32 and 103 kPa. How else can you assume a linear regression?

Line 174: Error in the reference.

Reviewer #2 (Remarks to the Author):

This paper presents a novel ex-vivo model demonstrating the ability to monitor the response of skeletal muscle to external mechanical stresses. A set of constant pressures were applied for various durations. It was demonstrated that increased duration and increased pressure showed a larger amount of damage to the skeletal muscle. The methods presented in the paper, with the use of a well designed load fixture as well as the assessment of the muscle using traditional H&E staining and Procion yellow staining show have generated valuable results. The comparison of the ex-vivo results presented in this paper to the in-vivo results of Linder-Gantz leads additional credibility to the model that was developed.

This study is a valuable addition to the literature.

The statistics performed (mean, SSE) are appropriate for the analysis.

Please check references in the paper. Missing reference in line 174.

Response to Reviewers

We are very grateful for the reviews provided by the editor and external reviewers. The comments are encouraging, and the reviewers appear to share our judgement that this study and its results are important for future research endeavours regarding the effect of mechanical loading on skeletal muscle.

Please see below, in blue, our detailed response to comments made by the reviewers. All line numbers refer to the manuscript file with changes.

Reviewer #1:

General comments:

1. The study is logically structured and comprehensible. However, I would like to see a more detailed overview of the animals used and the muscles taken from them. This concerns in particular the morphometric data regarding the animals, such as sex, weight, age.

The reviewer made a valid point of the importance of morphometric data regarding the animals and muscle specimen used in the study. We would like to point at Lines 69 – 71 where we state that we isolated the muscles *“from the hindlimbs of 6 – 7 weeks old male Sprague-Dawley rats [...]. The muscles had an average weight of 0.11 g (SOL) and 0.12 g (EDL). Their cross-sectional diameter was approximately 2 mm.”*

2. In general, I would also like to see more of the original data provided.

To keep our paper concise, we selected appropriate, representative data to include in the manuscript. However, additional original data will be available from the authors upon request.

Specific comments:

3. Line 71: Please specify SD's.

Please find the standard deviations for the average weight of the muscles in line 71: *“The muscles had an average weight of 0.11 ± 0.02 g (SOL) and 0.12 ± 0.02 g (EDL).”*

4. Line 79: Please introduce MOPS.

Please find the definition for MOPS added to line 79-80: *“The physiological buffer was prepared from 145 mM sodium chloride, 2 mM MOPS (3-(N-morpholino) propanesulfonic acid), [...].”*

5. Line 120: Here you use the Ogden model. Would there not be a material, that takes the anisometry of the musculature into account, more suitable? <https://doi.org/10.1016/j.ijsolstr.2015.11.008>

We agree with the reviewer that skeletal muscle tissue exhibits complex material behaviours, including viscoelasticity and anisometry, which are inherently difficult to fully represent in FE models. We chose to use the Ogden model with Prony-series expansion introduced by Bosboom et al. for two main reasons:

Firstly, unlike the approach mentioned by the reviewer, Bosboom et al.'s model is based on experimental results from transverse mechanical compression of rat skeletal muscle tissue, making it very relevant for our study where we also transversely compressed rat skeletal muscle tissue.

Secondly, the same model has been used by Linder-Ganz et al. to predict the internal loading conditions in their *in vivo* experiments, which served as a validation for our new *ex vivo* model. By using the same FE model, we therefore ensured better comparability between our study results.

To clarify this for the reader, we added the following sentences (line 122ff):

"The muscle model is based on experimental results from transverse mechanical compression of rat skeletal muscle tissue, making it very relevant to this study. Additionally, Linder-Ganz et al. [17] used the same model to predict the internal loading conditions in compressed skeletal muscle in their *in vivo* experiments, which served as validation for our new *ex vivo* model. By using the same FE model, we ensured better comparability between both studies."

However, as we mention in our discussion, we fully recognise the anisotropic nature of skeletal muscle tissue (line 392 ff) and suggest that a multi-scale FE model (line 443 ff) would be more suitable to assess the mechanical environment from the cell through to the tissue level.

6. Fig.2d: The graph should be filled with data points between 32 and 103 kPa. How else can you assume a linear regression?

The reviewer made a valid suggestion to add additional data points between 32 and 103kPa to verify the linear regression assumption we made. We therefore ran further simulations covering this loading range. The adjusted graph has been added to the manuscript to replace Fig 2d and can be seen below. Any descriptions relating to the graph have been changed accordingly (lines 136, 151).

6. Line 174: Error in the reference.

Thank you to the reviewer for spotting this mistake. This is an internal reference referring to Figure 2c of the manuscript. The reference was adjusted accordingly to display as “*The static compressive stress ranged from 3 – 68 kPa and was applied for 30 min – 120 min (Figure 2c).*”

Reviewer #2 (Remarks to the Author):

This paper presents a novel ex-vivo model demonstrating the ability to monitor the response of skeletal muscle to external mechanical stresses. A set of constant pressures were applied for various durations. It was demonstrated that increased duration and increased pressure showed a larger amount of damage to the skeletal muscle. The methods presented in the paper, with the use of a well designed load fixture as well as the assessment of the muscle using traditional H&E staining and Procion yellow staining show have generated valuable results. The comparison of the ex-vivo results presented in this paper to the in-vivo results of Linder-Gantz leads additional credibility to the model that was developed.

This study is a valuable addition to the literature.

The statistics performed (mean, SSE) are appropriate for the analysis.

1. Please check references in the paper. Missing reference in line 174.

Please see the response to comment 6 by reviewer 1. The reference has been adjusted accordingly.

General:

During the revision process, we noticed that the information we provided in the methodology section about the imaging equipment we used could be improved. We therefore added the following information to lines 194 - 196:

*“Fluorescently stained slides were imaged on a Leica **Microsystems SP8 confocal microscope in** widefield-mode. The system consisted of a Leica DMI8 inverted microscope **with images acquired via a 10x objective (HC PL APO CS2, NA 0.4)** and a DFC7000T CCD camera. For each mechanically damaged and control sample, $n = 6$ cross-sections were imaged with 10x magnification through an FITC filter (Ex 482/35, Em 536/40), followed by a Y5 filter (Ex 620/60 nm, Em 700/75 nm). Full cross-sectional images were composed through automated tiling with the Leica LAS X software (Leica Application Suite X V.3.1.5.16308).”*